# A New Low-Cost Internet of Things-Based Monitoring System Design for Stand-Alone Solar Photovoltaic Plant and Power Estimation

Batıkan Erdem Demir 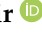

Department of Mechatronics Engineering, Faculty of Technology, Karabük University, 78050 Karabük, Türkiye; bedemir@karabuk.edu.tr

**Abstract:** The increasing demand for solar photovoltaic systems that generate electricity from sunlight stems from their clean and renewable nature. These systems are often deployed in remote areas far from urban centers, making the remote monitoring and early prediction of potential issues in these systems significant areas of research. The objective here is to identify maintenance requirements early and predict potential problems within the system. In this study, a cost-effective Internet of Things-based remote monitoring system for solar photovoltaic energy systems is presented, along with a machine learning-based photovoltaic power estimator. An Internet of Things-compatible data logger developed for this system gathers critical data from the photovoltaic system and transmits them to a server. Real-time visualization of these data is facilitated through web and mobile monitoring interfaces. The measured data encompass current, voltage, and temperature information originating from the photovoltaic generator and battery, alongside environmental parameters such as temperature, radiation, humidity, and pressure. Subsequently, these acquired data are employed for photovoltaic power estimation using machine learning techniques. This enables the estimation of potential issues within the photovoltaic system. In the event of a problem occurring within the photovoltaic system, users are alerted through a mobile application. Early detection and intervention assist in preventing power loss and damage to system components. When evaluating the results according to performance assessment criteria, it was observed that the random forests algorithm yielded the best results with an accuracy rate of 87% among the machine learning methods such as linear regression, support vector machine, decision trees, random forests, and k-nearest neighbor. When prediction models using other algorithms were ranked in terms of success, decision trees exhibited an accuracy rate of 81%, k-nearest neighbor achieved 79%, support vector machine reached 67%, and linear regression achieved 64% accuracy. In conclusion, the developed monitoring and estimation system, when integrated with web and mobile interfaces, has been demonstrated to be suitable for large-scale photovoltaic energy systems.

**Keywords:** solar photovoltaic power; low-cost; remote monitoring; Internet of Things; power estimation; machine learning

## 1. Introduction

The awareness regarding the utilization of renewable energy sources in energy production has been steadily increasing worldwide due to factors such as the rising global energy demand, the depletion of fossil fuel reserves, and the adverse effects of $CO_2$ emissions. Among the most widely employed renewable energy sources, solar energy is harnessed through solar panels to convert sunlight into electricity in photovoltaic (PV) systems. The annual average solar irradiance in Turkey is 1527.46 $kWh/m^2/year$, with an average sunshine duration of 2741.07 h, approximately equivalent to 27 million TOE (Tons of Oil Equivalent). As of the end of December 2022, the installed capacity of solar power plants in Turkey reached 9425.4 MW, indicating a growth of 1609.8 MW compared to the previous

year [1]. This global increase in solar energy installation capacity, observed in Turkey, brings about the necessity to address aspects such as power control, optimal energy generation, mitigation of power losses, power estimation, and maintenance and repair requirements in solar energy systems.

In this context, the conducted study comprises two fundamental parts. In the first part, a data recording and monitoring system is designed. Particularly crucial is the development of an automation system for remote monitoring of large-scale PV energy systems situated in remote areas, enabling early intervention against potential power losses. Considering the aforementioned attributes, this study designs an Internet of Things (IoT)-based data recording and monitoring system to real-time record and monitor parameters obtained from the PV panels, batteries, and energy system within their operational environment. The gathered data are stored in a database and can be tracked in real-time using the designed web and mobile interfaces. The data are presented both numerically and graphically to the user. Furthermore, the implemented monitoring system sends alert messages to the user in the event of a malfunction in the energy system, facilitating informed actions to prevent potential power loss through early intervention.

The second part of the study involves power forecasting using the collected data. The power output from PV panels varies based on factors such as geographical location, seasonal changes, and environmental conditions. Accurate power forecasting is essential for the efficient and economical utilization of solar panels as a reliable energy source. This enables the installation of controllable PV energy systems, guides electric companies, manages energy, optimizes energy levels, and identifies necessary panel adaptations to reach maximum production capacity. Moreover, it holds significant importance in terms of time savings and reduction of additional labor costs. Therefore, the estimation of power output values and load trends for renewable power facilities like PV energy systems emerges as a fundamental process [2].

Presently, the prevalent approach for power forecasting in PV energy systems involves analyzing historical data and considering seasonal, daily, and hourly variations to predict future power generation. Artificial neural networks and regression models are among the methods employed for this purpose [3]. In this study, power forecasting is conducted using the data obtained from Karabük province in Turkey's Western Black Sea region, where the annual sunshine duration is 2402 h, and the annual radiation value is 1369 kW/h per square meter. Meteorological data, such as humidity, temperature, pressure, and time information, are utilized in power forecasting through machine learning techniques [4]. To determine the most successful machine learning method for power forecasting, linear regression, support vector machines (SVM), decision trees, random forests, and k-nearest neighbors (KNN) algorithms are sequentially employed. The results obtained from each algorithm are presented in a comparative manner.

The subsequent section provides an in-depth review of the relevant literature on the topic. Section 3 details the experimental setup, dataset, and algorithms employed. In Section 4, experimental studies and the resulting findings are thoroughly analyzed, and in Section 5, the conclusions are discussed.

## 2. Related Work

The first part of the literature review addresses the studies related to monitoring of PV energy systems. In the initial research efforts concerning the monitoring of PV energy systems, wired systems utilizing RS232 and RS485 communication protocols were employed for data transmission [5,6]. Due to exposure to environmental factors, such as rain, temperature, and humidity, the cables carrying data in these systems necessitated additional maintenance costs. In contrast, wireless monitoring systems are less affected by environmental conditions compared to wired monitoring systems and, especially in real-time applications, they possess quicker decision-making capability. Additionally, they convey information over a longer range with higher accuracy. In their work, Rouibah et al. [7] developed a low-cost IoT-based tracking system for maximum power point track-

ing (MPPT) in PV systems. Deshmukh and Bhuyar [8] addressed the automation of solar PV power generation. An IoT platform was utilized to monitor and control solar energy production. Cheddadi et al. [9] aimed to provide a cost-effective and open-source IoT solution using the ESP32 board to intelligently gather and real-time monitor the generated power and environmental conditions of solar stations. Adhya and co-workers [10] discussed an IoT-based, low-cost monitoring system for solar PV installations. Luwes and Lubbe [11] developed an IoT device that individually monitored each PV array and provided feedback on their efficiencies to prevent power losses in large solar farms. Lee et al. [12] described an IoT-based software architecture for continuous monitoring of solar panel efficiency. López-Vargas et al. [13] introduced IoT-based application innovation to a low-cost Arduino microcontroller-based solar data logger. Fernandez et al. [14] proposed a fully open-source software-based IoT solution for monitoring PV installations. Gupta et al. [15] systematically presented all design stages of a low-cost IoT-based data collection system. In Nurhafizah's study [16], an IoT-based real-time monitoring system for renewable stand-alone power plants was discussed. Portalo et al. [17] presented an open-source hardware and software-based monitoring system for tracking the temperature of PV generators in smart microgrids. The monitoring system they developed utilized an Arduino microcontroller and a Raspberry Pi microcomputer. Boubakr et al. [18] assessed a remote monitoring system for photovoltaic power generation stations using IoT and a state-of-the-art tool for virtual supervision.

Continuing the literature review, the second part of the study examines the power forecasting methods used in solar PV systems. Lorenz et al. [19] presented a comparative study of solar irradiance predictions obtained through multiple linear regression methods and Artificial Neural Networks (ANN) models, revealing PV panel power output characteristics using weather data. Wang and colleagues [20], apart from the aforementioned studies, concluded that ANN is the most suitable method for predicting PV power outputs. Shi et al. [21] conducted research using Support Vector Machines (SVM), a machine learning approach, to predict PV system power outputs. Kou et al. [22] utilized a backpropagation-trained ANN structure along with meteorological data to forecast solar panel output power. Zhang et al. [23] hybridized the Particle Swarm Optimization (PSO) evolutionary algorithm with an ANN, incorporating irradiance values as inputs, to obtain solar radiation prediction in their training approach. Qasrawi and Awad [24] designed a multi-layered feedforward ANN using panel outputs from differently located solar panels along with satellite data. Zhu et al. [25] employed wavelet transform to extract useful information from complex PV output power data and constructed an ANN model.

In the studies by Paulin and Praynlin [26], a comparative investigation was presented using a backpropagation-based Artificial Neural Network (ANN) where inputs encompassed average ambient temperature, average panel temperature, average inverter temperature, solar irradiance, and wind speed data, while the output consisted of power data. Rana et al. [27] offered a comprehensive evaluation of a series of leading methods to forecast solar power output profiles one day in advance. Kwon et al. [28] proposed the Naive Bayes (NB) classification method, employing publicly available outdoor data (temperature, humidity, dew point, and sky coverage) for solar irradiance prediction. Dinçer and İlhan [29] comparatively employed feedforward-backpropagation artificial neural networks and KNN algorithms using temperature, humidity, pressure, and irradiance values to predict output power of PV panels. Gumar and Demir [30] utilized metaheuristic algorithms such as Genetic Algorithm (GA), PSO, and Artificial Bee Colony (ABC) in conjunction with an ANN model to predict solar energy outputs.

In this study, current technologies were integrated to enable real-time monitoring of the electrical data from the IoT-based PV energy system and meteorological data from the environment accessible, both via the internet and through the developed mobile application. Subsequently, power forecasting of the generated energy in the PV system was conducted using machine learning methods based on these data. The combination of the cost-effective IoT-based real-time monitoring system and machine learning techniques for power forecast-

ing represents a significant contribution and innovative aspect of this study to the literature. The following section presents the developed IoT-based PV monitoring system.

## 3. Materials and Methods

### 3.1. Design of an IoT-Based Solar PV Data Monitoring System

In the designed system, real-time IoT-based condition monitoring of a solar PV panel and a battery charged by this panel is conducted. Issues occurring within the system are identified through the acquired data. Current, voltage, and temperature values of the panel and battery, along with environmental parameters such as humidity, temperature, and irradiance, are measured using relevant sensors. The obtained data are sent to SD cards and Firebase servers through the NodeMCU v2 Wi-Fi microcontroller board and stored. Firebase was chosen due to its ease of use and quick integration into the project. It provides real-time database connectivity, allowing multiple users to observe changes in data when they are created or edited. Despite not being open-source, Firebase offers open-source libraries and SDKs for developers. Data tracking can be carried out in real-time through Firebase, as well as via a mobile application. The mobile application has been developed in the Flutter environment using the Dart programming language. Moreover, various potential errors in the system can also be monitored in real-time through these systems. These errors encompass low panel voltage, low panel current, high panel temperature, low battery voltage, high battery temperature, and PV panel connection fault. The block diagram of the developed IoT-based PV data recording and monitoring system is provided in Figure 1.

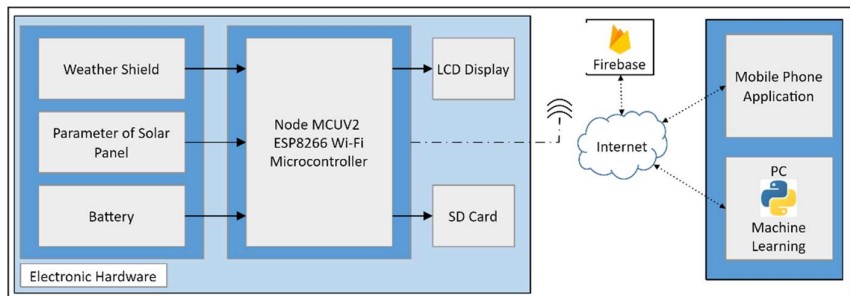

**Figure 1.** Block diagram of the proposed monitoring and estimation system.

The NodeMCU v2 Wi-Fi microcontroller board, frequently favored in IoT applications, renders the system controllable and monitorable from any location worldwide. The development board possesses features such as a 10-bit ADC, USB-TTL converter, 17 GPIO pins, Wi-Fi module for wireless network connectivity, and ease of programming. In this study, various sensors are utilized to measure values requiring assessment, including PV panel voltage, PV panel current, PV panel temperature, battery voltage, battery current, and battery temperature, as well as humidity, temperature, and irradiance values of the ambient air. The ACS712-30A current sensor is employed for measuring the current of both the PV panel and the battery. The LM35 temperature sensor is used for measuring the temperature of the PV panel and the battery. The Sparkfun weather shield is employed for reading temperature, humidity, pressure, and irradiance data from the air. Simultaneously, the GP-735 GPS (Global Positioning System) sensor on the board provides location information. The integration of the weather measurement and microcontroller Wi-Fi boards has resulted in the creation of an electronic board for the IoT-based data acquisition system. The measurement setup, situated within the premises of Karabük University, is depicted in Figure 2.

The NodeMCU v2 module within the solar PV data measurement, recording, and monitoring apparatus should initially establish a connection with a wireless network. Once connected, data from the sensors are sequentially measured, displayed on the built-in LCD display, saved to an SD card, and sent to the Firebase real-time database. The Firebase cloud

system is a platform that enables the creation of mobile and web applications for monitoring real-time data, maintaining records and session information, making new announcements, and creating control units. It can be used freely for these purposes. The solar PV data sent to the cloud system can be tracked by end-users via smartphones through the developed mobile application. Additionally, the analysis and visualization of these data in a web-based manner are facilitated through the ThingSpeak IoT platform, where we direct the data flow in the cloud environment. A comparison between the proposed IoT-supported data monitoring system and some existing data monitoring systems is provided in Table 1.

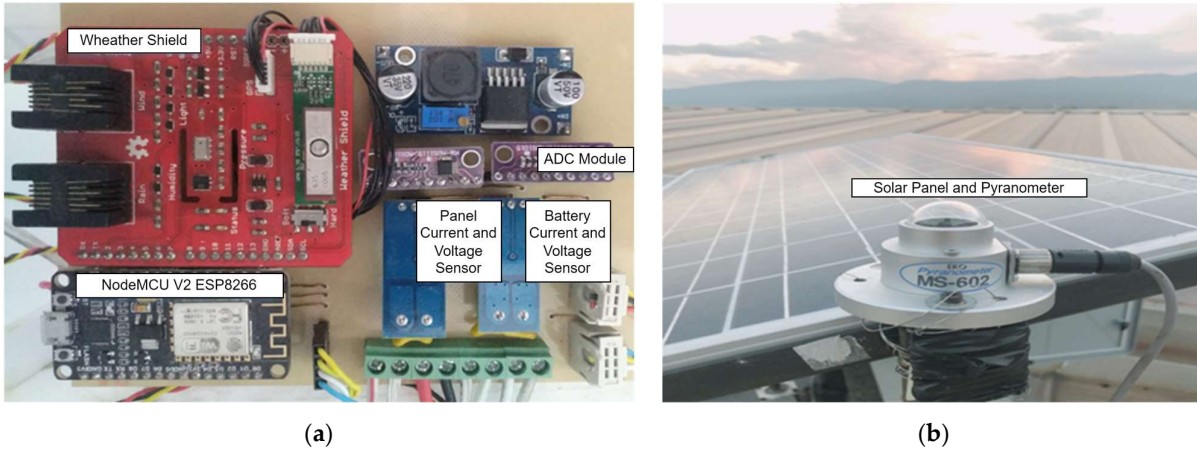

(**a**)     (**b**)

**Figure 2.** (**a**) IoT-based data acquisition board; (**b**) Experimental setup.

**Table 1.** Comparison between the proposed solar PV monitoring system and existing solar PV monitoring systems.

| Author | Network | Hardware | Software | Cost (€) |
|---|---|---|---|---|
| Koutroulis and Kalaitzakis [6] | Wired | NI PCI-6024E DAQ | LabView 6.1 | 749 € |
| Chouder et al. [31] | Wired | Agilent 34970A DAQ | LabView 2011 | 1420 € |
| Ferdoush and Li [32] | Wireless | Raspberry Pi and Arduino UNO | Arduino IDE 1.0 | 60 € |
| Rezk et al. [33] | Wired | NI USB-6009 DAQ | LabView 2016 | 120 € |
| Proposed IoT-Based System | Wireless | ESP8266 | Arduino IDE 2.2 | 5 € |

In this study, following the integration of IoT-based electronic hardware, the ThingSpeak platform, and the mobile application, experimental measurements were conducted. The data from the experimental measurements were recorded and monitored through internet and mobile platforms. The prototype system, which was purposefully designed and cost-effective, utilized a NodeMCU v2 microcontroller board, a Sparkfun WeatherShield weather measurement board, an ADS1115 ADC module, an ACS712-30A current sensor, an LCD display, an LM35 temperature sensor, and a Sparkfun (SparkFun Electronics, Boulder, CO, USA) GP-735 GPS receiver. The approximate budget spent on creating this prototype system was around 120 €. Examining the results obtained from the testing process, it was observed that the developed system accurately measured and transmitted data. Electrical and meteorological measurements were corroborated through comparison with diverse sources. The mentioned measurements in the study were repeated and recorded at five-minute intervals. Daily variation graphs for the output voltage of the monitored PV panel, battery voltage, daily irradiance, and ambient temperature are provided in Figure 3.

Figure 4 illustrates the visualization of the data transferred to the Firebase real-time database and sent to the cloud platform, where they are displayed via the developed mobile application.

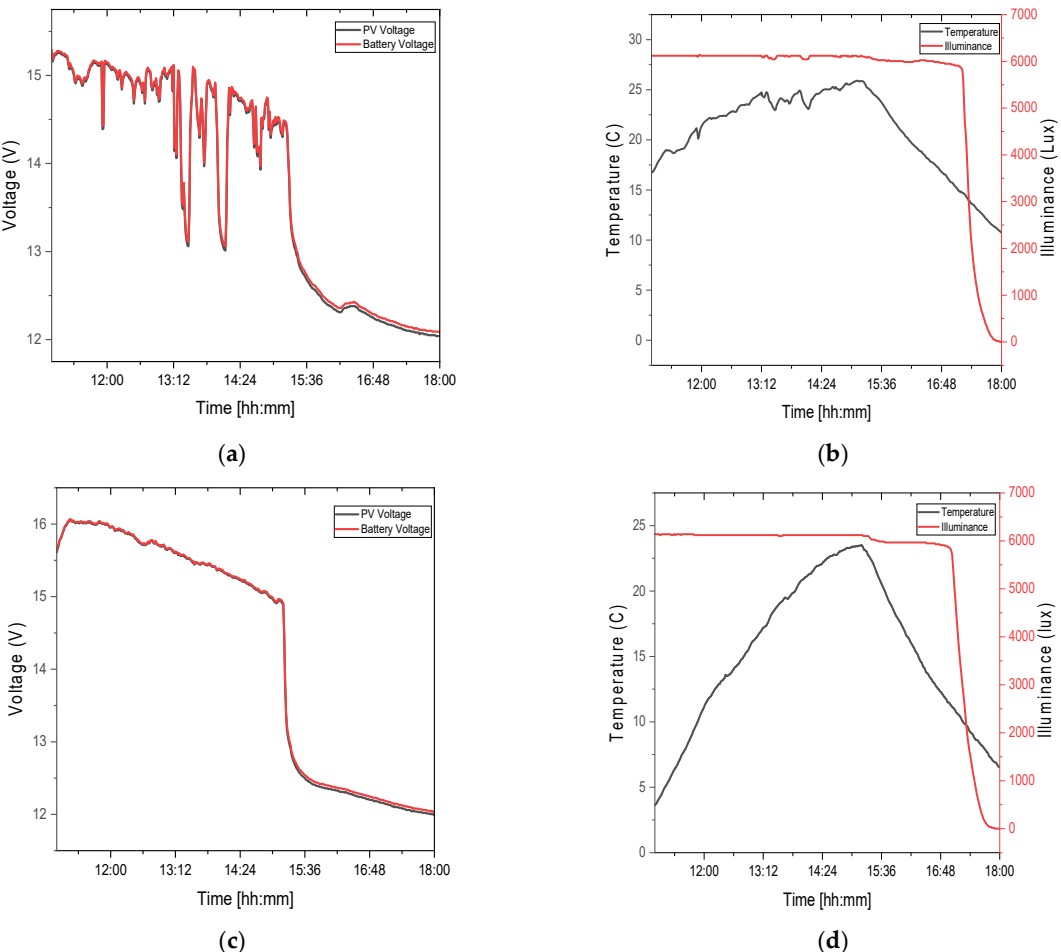

**Figure 3.** Voltage, illuminance, and ambient temperature measured in (**a**,**b**) Karabuk (9 November 2021) and (**c**,**d**) Karabuk (16 November 2021).

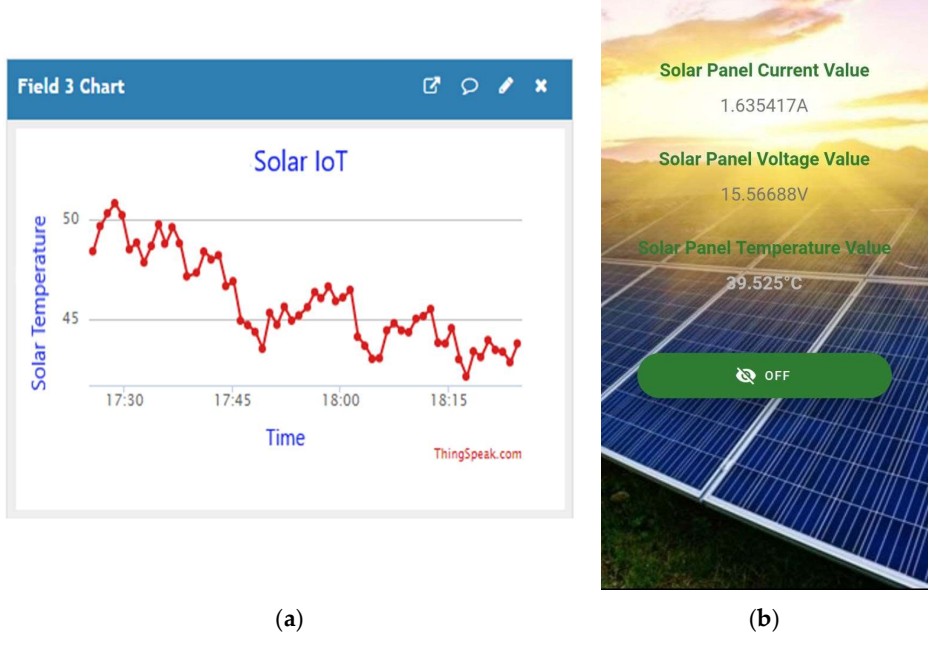

**Figure 4.** Visualization of solar data using (**a**) the ThingSpeak server and (**b**) the developed mobile application.

The subsequent section addresses the analysis of the collected data with the aim of utilizing machine learning methods to predict the future performance of the solar PV system.

### 3.2. Solar PV Power Estimation using Machine Learning Methods

For solar PV power estimation, we have a dataset containing time, temperature, pressure, and humidity data. This dataset is based on five-minute weather measurements. In order to investigate the relationship between solar energy and meteorological data, certain weather parameters have been collected, aiming to accurately predict PV power generation. The steps related to this part of the study involve acquiring and preprocessing the dataset, followed by splitting the data into training and testing sets, applying classification techniques, and making predictions based on the results. Data preprocessing is necessary to clean the data and prepare them for the utilization of various learning models, thus enhancing accuracy and efficiency. Training and testing data are separated from the preprocessed data. The model is trained using the training data and its predictions are verified using the testing data. Data splitting generally refers to dividing the available data into two parts for cross-validation purposes. The first dataset is used to build a prediction model, while the second dataset is used to evaluate the model's performance. The training percentage is set at 80%, while the testing percentage is set at 20%. Solar PV power generation is predicted using machine learning methods such as linear regression, SVM, decision trees, random forests, and KNN, as proposed in the article. Linear regression is one of the fundamental and commonly used regression methods [34]. Linear prediction functions are used to represent the relationship between input and output variables, and the method of least squares is employed to estimate unknown model parameters from the data. An iterative method, such as a series of linear equations or gradient descent, can be used to estimate parameter values. In the study, a scaling process was applied to standardize the input data, followed by feature provisioning and then scaling for feature standardization. The sensitivity of SVM depends on the kernel function and other variables. The Grid Search approach was employed to find optimal settings. Methods such as decision trees and random forests are commonly used in various data science challenges. The random forests method is a tree-based machine learning approach that can be used for regression and classification. It also conducts dimensionality reduction, checks for missing and abnormal values, and performs various additional data exploration activities. The bagging approach is used to train random forests. This method allows for the use of a large number of examples during training as the dataset is sampled with replacement. The KNN algorithm is a distance-based classifier extensively utilized in artificial intelligence, especially in pattern recognition. In KNN-based classification, distances between training and test data are calculated to select the nearest K samples to the test example. Subsequently, the class of the test example is determined through majority voting based on the class information of the selected K samples [35].

In this study, the prediction models mentioned above were constructed using the scikit-learn library in Python to predict solar PV power output based on multiple meteorological parameters. Subsequently, a comparative performance analysis is necessary to compare the prediction results of the created models and determine the most accurate model based on specific evaluation criteria. To evaluate the performance of the prediction model, one or more evaluation methods may be preferred. In this study, metrics such as Mean Absolute Error (MAE), Mean Squared Error (MSE), Root Mean Squared Error (RMSE), and R-squared ($R^2$) values were calculated to evaluate the performance of the models on the test data. MAE is a metric used to measure how close the predicted values are to the measured values. MSE measures the average of the squared errors, thus encompassing both how widely the predictions are spread from the actual samples and how far the mean estimated value deviates from the true value. RMSE is calculated to assess the accuracy of a specific approach's predictions and indicates the scattering level generated by the model. R-squared is used to indicate how close the prediction model results are to the actual measured data

line, known as a fitted regression line. For higher modeling accuracy, the MAE, MSE, and RMSE indices should be closer to zero, while the $R^2$ value should be closer to 1. The equations for MAE, MSE, RMSE, and $R^2$ performance measurement methods are given as Equation (1)–(4), respectively.

$$\text{MAE} = \frac{1}{n}\sum_{i=1}^{n}|y_i - \hat{y}| \tag{1}$$

$$\text{MSE} = \frac{1}{n}\sum_{i=1}^{n}(y_i - \hat{y})^2 \tag{2}$$

$$\text{RMSE} = \sqrt{\frac{1}{n}\sum_{i=1}^{n}(y_i - \hat{y})^2} \tag{3}$$

$$\text{R}^2 = 1 - \frac{\sum(y_i - \hat{y})^2}{\sum(y_i - \overline{y})^2} \tag{4}$$

Here, $n$ represents the number of data points, $y$ is the true value, $\hat{y}$ is the predicted value, and $\overline{y}$ is the mean value. The comparison of the performance of the prediction models based on these performance measurement criteria is presented in the results section.

## 4. Result and Discussion

In order to comprehensively observe the solar PV power generation process based on meteorological data, a matrix consisting of correlation coefficient pairs was constructed. This enabled the identification of collinearity between the existing features and the power output. The data used in this study were obtained from a research project numbered KBU-21-DS-018 and titled "IoT-Based Condition Monitoring and Fault Analysis of Solar Panels". Figure 5 provides the correlation between the existing features and solar PV power output.

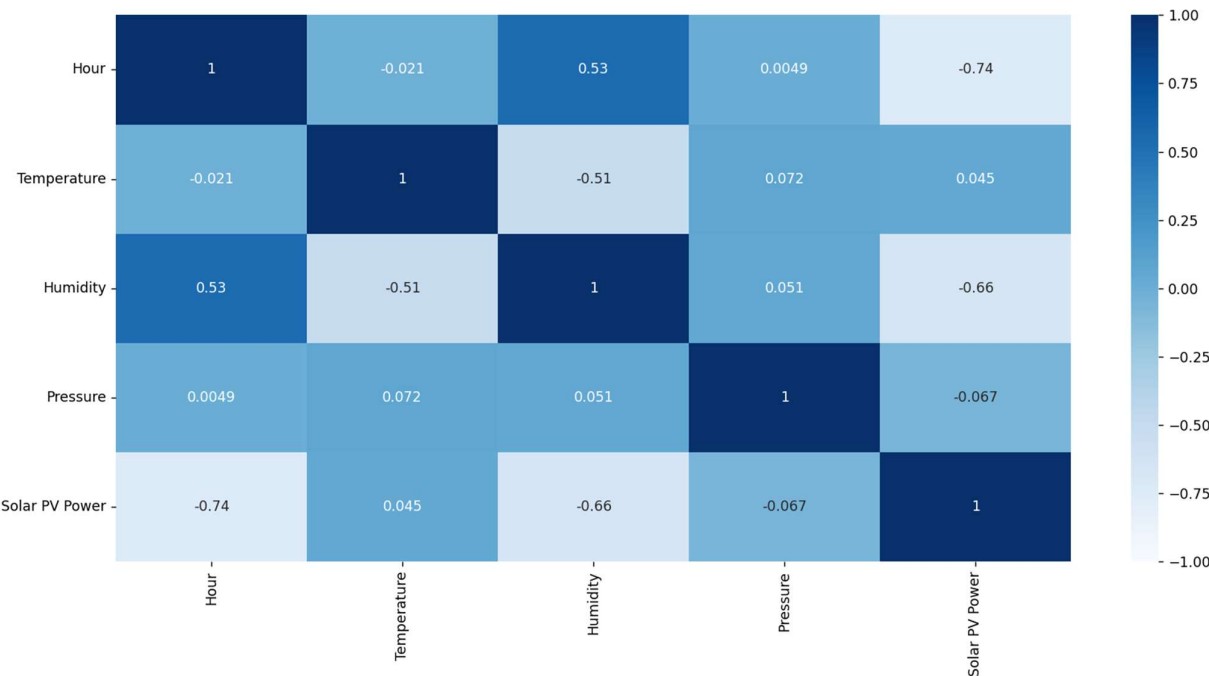

**Figure 5.** Correlation between current features and solar PV power output.

From the correlation graph, it can be observed that the features with the highest correlation to power output are hour and humidity. Additionally, a similar strong correlation between humidity and temperature is also evident. Following the identification of the relationship between meteorological data and solar PV power output, the dataset was

subjected to machine learning models. The graph in Figure 6 presents the predicted values against the actual test data for solar PV power output estimations performed using linear regression, SVM, decision trees, random forest, and KNN algorithms.

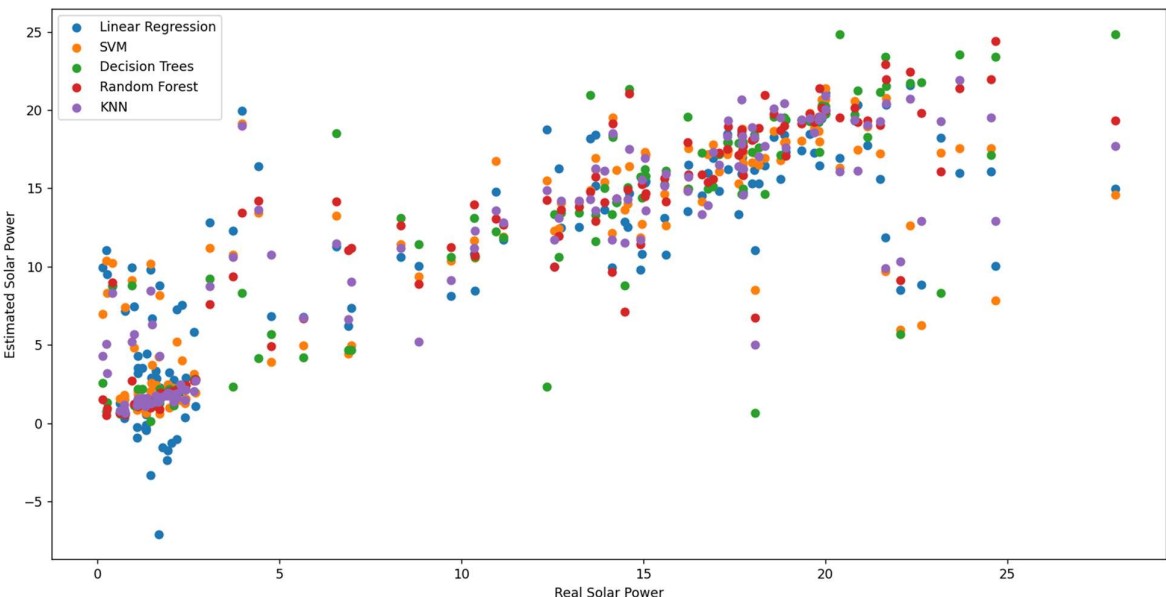

**Figure 6.** Predicted values against real test data.

After obtaining the graph depicting predicted values corresponding to real test data, the performance of the models was quantified numerically. At this stage, metrics such as MAE, MSE, RMSE, and $R^2$ were employed as performance evaluation criteria. The comparison of model performances based on these evaluation criteria is presented in Table 2.

**Table 2.** Comparison of prediction models based on MAE, MSE, RMSE, and $R^2$ evaluation criteria.

| Model | MAE | MSE | RMSE | $R^2$ |
|---|---|---|---|---|
| Linear Regression | 3.41 | 23.53 | 4.85 | 0.64 |
| Linear Regression-index | 2.24 | 2.74 | 1.66 | 1.35 |
| SVM | 2.76 | 21.25 | 4.60 | 0.67 |
| SVM-index | 1.81 | 2.47 | 1.57 | 1.29 |
| Decision Trees | 1.72 | 12.48 | 3.53 | 0.81 |
| Decision Trees-index | 1.13 | 1.45 | 1.20 | 1.07 |
| Random Forest | 1.52 | 8.57 | 2.92 | 0.87 |
| Random Forest-index | 1.00 | 1.00 | 1.00 | 1.00 |
| KNN | 2.15 | 13.48 | 3.67 | 0.79 |
| KNN-index | 1.41 | 1.57 | 1.25 | 1.10 |

According to the performance evaluation criteria presented in Table 2, it can be deduced that the most successful prediction model across all domains is the random forest, while the least performing prediction model is linear regression. Based on the MAE evaluation criterion, the random forest algorithm exhibits approximately 13% less error than the decision trees algorithm, which provides the closest result. The KNN algorithm is the third one, and it has 41% more error than the random forest algorithm. The error difference between the SVM, linear regression algorithms, and the random forest algorithm is 1.81 times, and 2.24 times, respectively.

In terms of the MSE evaluation criterion, the random forest algorithm outperforms the decision trees algorithm, which provides the closest result, by approximately 45%. The KNN algorithm is the third one, and it has 57% more error than the random forest

algorithm. The error difference between the SVM, linear regression algorithms, and the random forest algorithm is 2.47 times, and 2.74 times, respectively.

Considering the RMSE evaluation criterion, the random forest algorithm possesses about 20% less error than the decision trees algorithm, which provides the closest result. The error difference between the KNN, SVM, and linear regression algorithms, and the random forest algorithm is 25%, 57%, and 66%, respectively.

With regards to the $R^2$ evaluation criterion, the random forest algorithm performs about 7% better than the decision trees algorithm, which provides the closest result. The difference in performance between the KNN, SVM, and linear regression algorithms, and the random forest algorithm is 10%, 29%, and 35%, respectively.

According to the accuracy performance comparison of machine learning methods on test data, the prediction model utilizing the random forest algorithm has yielded the best result with an accuracy rate of 87%. When the prediction models using other algorithms are ranked in terms of success, decision trees have an accuracy rate of 81%, KNN has 79%, SVM has 67%, and linear regression has 64% accuracy rate.

## 5. Conclusions

In the first phase of this study, a reliable and cost-effective data monitoring system was developed to enable the remote and real-time monitoring of solar PV energy systems. The infrastructure of the IoT-based system consists of a data recording unit, a cloud system, a web interface, and a mobile application. Data from the solar PV system can be monitored independently of location through a mobile application that can be installed on smartphones or via the ThingSpeak platform. The distinctiveness of the proposed system at this point lies in making solar PV systems and the weather parameters affecting them accessible in a cost-effective manner through open-source software and mobile applications. In the second phase of the study, prediction models were developed to determine the expected data for comparison with the obtained real measurement data. This approach allows for the identification of potential system issues based on the difference between actual solar PV power and expected solar PV power derived from existing meteorological data. Machine learning methods including linear regression, SVM, decision trees, random forests, and KNN were employed to develop prediction models based on measurement data. The performance of these models was then numerically compared using performance metrics including MAE, MSE, RMSE, and $R^2$. Through the comparison conducted using these performance metrics, the random forests algorithm emerged as the most successful model across all criteria. In terms of accuracy performance on test data, the random forests algorithm achieved the highest accuracy rate of 87%. Other algorithms ranked in descending order of success were decision trees with 81%, KNN with 79%, SVM with 67%, and linear regression with 64% accuracy rates. In conclusion, this study successfully presents a cost-effective solar PV power monitoring system and a machine learning-based solar PV power predictor.

In future work, increasing the number of parameters in the dataset, experimenting with different machine learning methods, and testing the developed system in large-scale solar fields are planned.

**Funding:** This research was funded by Karabük University within the scope of Scientific Research Projects, grant number KBU-21-DS-018.

**Institutional Review Board Statement:** Not applicable.

**Informed Consent Statement:** Not applicable.

**Data Availability Statement:** Not applicable.

**Conflicts of Interest:** The author declares no conflict of interest.

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
