# Peer review of "A New Low-Cost Internet of Things-Based Monitoring System Design for Stand-Alone Solar Photovoltaic Plant and Power Estimation"

_applsci, doi:10.3390/app132413072_

Round 1
Reviewer 1 Report
Comments and Suggestions for Authors
1-Please remove all the abbreviations in the abstract and define them for the first time in the introduction section and after that.
2- Add some numerical results to the abstract.
3- Use shorter titles for each section such as 3.1. and etc.
4- Use the English language for the text in all figures such as Fig. 2.
5- The quality of figures in some cases is low such as Fig. 4.
6- Add future work suggestions to the conclusion.
7- What are the time and space complexity of your model?
8- A sensitivity analysis is required on the model. because I think the results in Fig. 7 are not accurate and tuning of the input parameters of the algorithms changes the results.
Comments on the Quality of English Language1- The language of the paper should be checked by a native researcher to improve the paper quality and flow of reading.
Reviewer 2 Report
Comments and Suggestions for Authors
1. The abstract is not concise, and the text is sloppy. The problem, motivation, research gap, and the concluding results are not addressed in the abstract. It must be written again, carefully.
2. The components described in Fig. 2 should be rewritten in English.
3. Comparing with the literature, the added value of this paper is not clear. The comparison made in Table 1 should include all aspects and criteria that makes the proposed system more attractive.(table 1 should show more quantitative and qualitative comparison)
4. Why did the authors use 4 metrics mentioned in equation 1 through equation 4?
5. What are the differences between these metrics?
6. The sentence in conclusion “It exhibited superiority over other algorithms in the range of 13% to 124% for MAE, 45% to 174% for MSE, 25% to 66% for RMSE, and 7% to 35% for R2.” Does not conclude clear results due to the accuracy of metrics results.
This section should be explained and discussed in a way that clear results are attained.
The big range 13-124%. Why is it that big?
7. The authors claimed that the proposed system is cost-effective without providing any comparison. The price of ESP chip is more expensive than Arduino used in ref [24]
8. The results shown in Fig. 6 look interesting. However, if they can be grouped, clustered, or averaged, the results of the accuracy would be more realistic.
Comments on the Quality of English LanguageEnglish language requires polish and some sentences should be broken into two or three to ease the readability.
Reviewer 3 Report
Comments and Suggestions for Authors
The topic of monitoring photovoltaic facilities and low-cost technology fits the scope of the Journal. The manuscript requires extra efforts to improve its quality and presentation for the prestigious journal Applied Sciences. A set of comments are expounded hereafter.
- The manuscript is, in general, well organized and well written. However, there are some issues regarding the format of the document, as commented below.
The abbreviations should be defined the first time that they appear. For example, in the Abstract, PV, IoT, KNN are directly used.
In line 45, the abbreviation FV is used. Perhaps, it should be PV. The same issue occurs in lines 106, 155 – 160 and 222.
The section Authors contributions is missing.
- About the content of the manuscript, it covers a very interesting topic. The comments after a careful revision are the following:
A keyword to include would be “low-cost”, if the authors agree with the suggestion. Indeed, “open-source” could also be interesting.
In the first section, a common practice in scientific papers consists on including a paragraph at the end of the Introduction to briefly describe the structure of the rest of sections in the manuscript. It is suggested to add such a paragraph for a better readability.
The literature review conducted in the second section is well organized but it lacks some recent publications which also deal with low-cost IoT technology for monitoring PV panels. Some papers are now suggested for consideration of the authors in this regard:
- Enhancing Virtual Real-Time Monitoring of Photovoltaic Power Systems Based on the Internet of Things. Electronics 2022, 11, 2469. https://doi.org/10.3390/electronics11152469
- A Low-Cost IoT System for Real-Time Monitoring of Climatic Variables and Photovoltaic Generation for Smart Grid Application. Sensors 2021, https://doi.org/10.3390/s21093293
- Monitoring System for Tracking a PV Generator in an Experimental Smart Microgrid: An Open-Source Solution. Sustainability, 2021, https://doi.org/10.3390/su13158182
- Low-cost web-based Supervisory Control and Data Acquisition system for a microgrid testbed: A case study in design and implementation for academic and research applications. Heliyon 2019, DOI: 10.1016/j.heliyon.2019.e02474.
The authors use various components of open source nature, which is a positive feature. In fact, the authors mention that it is open source but do not expound the benefits that it involves like information in the Internet, inexpensive (or null) acquisition, etc. Therefore, it is strongly suggested to comment in a brief manner the advantages of open source technology. This can be conducted in section 3.1.
The symbol of the programming language Python can be observed in figure 1; however, it does not appear in the text. This must be solved to enhance the provided description as well as to avoid misleading to the readers.
The database Firebase is not open source, so it should also be mentioned. Moreover, it is not SQL database, an important feature that should also be commented for a comprehensive description. The reasons to choose such database could be briefly indicated. Have the authors considered using some open source database such as MySQL, etc.?
What is the software used to develop the app for mobile visualization of the gathered data?
Have the authors studied the operation of the developed system for mid- or long-term periods? This aspect is relevant in the field of low-cost systems.
The authors claim that the reported system is low-cost; however, there is no mention to the costs involved in its development. To address this issue, for example, a table with the costs of hardware and software could be added. In the case of the Arduino IDE software, the cost is null, but is should also be indicated.
A desirable comment in the Conclusions section deals with future works that the authors are considering on the view of their developed work. For instance, the aforementioned long-term operation could be considered.
Round 2
Reviewer 1 Report
Comments and Suggestions for Authors
No comment
Comments on the Quality of English LanguageNo comment
Reviewer 2 Report
Comments and Suggestions for Authors
Thanks for addressing my comments
Comments on the Quality of English Languagea bit more polish is required
Reviewer 3 Report
Comments and Suggestions for Authors
The new version has addressed the reviewer concerns in a proper manner.